# Prevalence of Variant GTR^I^
*Staphylococcus aureus* Isolated from Dairy Cow Milk Samples in the Alpine Grazing System of the Aosta Valley and Its Association with AMR and Virulence Profiles

**DOI:** 10.3390/antibiotics14040348

**Published:** 2025-03-27

**Authors:** Valentina Monistero, Delower Hossain, Sara Fusar Poli, Elizabeth Sampaio de Medeiros, Paola Cremonesi, Bianca Castiglioni, Filippo Biscarini, Hans Ulrich Graber, Giulia Mochettaz, Sandra Ganio, Alessandra Gazzola, Maria Filippa Addis, Claudio Roullet, Antonio Barberio, Silvia Deotto, Lara Biasio, Fernando Ulloa, Davide Galanti, Valerio Bronzo, Paolo Moroni

**Affiliations:** 1Dipartimento di Medicina Veterinaria e Scienze Animali, Università Degli Studi di Milano, 26900 Lodi, Italy; valentina.monistero@unimi.it (V.M.); delower.hossain@unimi.it (D.H.); sara.fusar@unimi.it (S.F.P.); elizabeth.medeiros@ufrpe.br (E.S.d.M.); filippa.addis@unimi.it (M.F.A.); davide.galanti@unimi.it (D.G.); valerio.bronzo@unimi.it (V.B.); 2Laboratorio di Malattie Infettive Degli Animali-MiLab, University of Milan, 26900 Lodi, Italy; 3Department of Medicine and Public Health, Faculty of Animal Science and Veterinary Medicine, Sher-e-Bangla Agricultural University (SAU), Dhaka 1207, Bangladesh; 4Laboratório de Inspeção de Carne e Leite, Departamento de Medicina Veterinária, Universidade Federal Rural de Pernambuco (UFRPE), Recife 51171-900, PE, Brazil; 5Institute of Agricultural Biology and Biotechnology, National Research Council, 26900 Lodi, Italy; paola.cremonesi@ibba.cnr.it (P.C.); bianca.castiglioni@ibba.cnr.it (B.C.); filippo.biscarini@ibba.cnr.it (F.B.); 6Food Microbial Systems, Microbiological Safety of Foods of Animal Origin Group, Agroscope, 3003 Bern, Switzerland; hans.graber-p@outlook.com; 7Dipartimento di Prevenzione AUSL Della Valle d’Aosta, 11100 Aosta, Italy; giulia.mochettaz@libero.it (G.M.); sganio@ausl.vda.it (S.G.); croullet@ausl.vda.it (C.R.); 8Istituto Zooprofilattico Sperimentale Della Lombardia e Dell’emilia-Romagna, 26900 Lodi, Italy; alessandra.gazzola@izsler.it; 9Istituto Zooprofilattico Sperimentale Delle Venezie, 35020 Legnaro, Italy; abarberio@izsvenezie.it (A.B.); sdeotto@izsvenezie.it (S.D.); lbiasio@izsvenezie.it (L.B.); 10Escuela de Graduados, Facultad de Ciencias Veterinarias, Universidad Austral de Chile, Valdivia 5090000, Chile; fernando.ulloa@uach.cl

**Keywords:** alpine grazing system, *Staphylococcus aureus*, intramammary infection, genotyping, antimicrobial resistance, virulence genes

## Abstract

**Background/Objectives**: In the Aosta Valley, the alpine grazing system integrates livestock production and land management. Valdostana breeding has adapted to this mountainous region, but the spread of *Staphylococcus aureus* within pastures may impact animal health. The aim of this study was to provide an overview of *S. aureus* genotypes associated with antimicrobial resistance (AMR) and virulence profiles in four dairy herds in the Aosta Valley from July 2022 to August 2023. **Methods**: A total of 468 composite milk samples were collected at three timepoints: T1 (pasture-livestock system), T2 (farm-livestock system), and T3 (pasture-livestock system). *S. aureus* isolates were characterized by antimicrobial susceptibility testing, ribosomal spacer (RS)-PCR, multilocus sequence typing (MLST), PCR analysis for 28 virulence genes and 6 AMR genes, and *adlb*-targeted real-time PCR. **Results**: RS-PCR analysis of 82 *S. aureus* strains revealed 12 genotypes (GT) in eight clusters (CL). The most prevalent variant was GTR^I^ (61%), followed by GTB (15%). Resistance to penicillin was high (69%), with CLR strains showing 88% resistance, and 51% resistance to amoxicillin plus clavulanate. All strains were susceptible to cephalosporins and oxacillin. Macrolide resistance was low (4%), and multi-drug resistance was 6%. AMR gene presence corresponded with susceptibility, with *blaZ* detected in 94% of CLR strains. CLR strains also possessed genes for biofilm formation and virulence factors. **Conclusions**: This study highlights the presence of AMR and virulence factors in *S. aureus* strains from alpine grazing systems, underscoring the need for ongoing monitoring to mitigate risks to animal health.

## 1. Introduction

The Aosta Valley is a mountainous region located in Northwestern Italy that is rich in vast pastures with the highest altitudes in the Alps, at 1600 m above sea level (asl), where dairy herds are typically led up during the summer months. The grazing season lasts approximately 100 days, traditionally starting in early June and ending on 29th September, coinciding with the feast of Saint Michael. This ancient practice represents an example of sustainable integration between livestock production and land management, highlighting the importance of the multifunctional role of mountain farming. The alpine grazing system offers ideal conditions to benefit fully from natural sources, as the very nutritious grass used for animal feed results in high-quality milk [1]. Valdostana represents a dual breed particularly adapted to this region, whose milk production results in mainly traditional dairy products such as Fontina, a semi-cooked cheese made from raw, full cream cow’s milk using one single milking of the native Valdostana breed. Due to its specific organoleptic properties and unique production technology, it received the Protected Designation of Origin (PDO) status from the European Communities in 1996. Despite the high environmental adaptability of this native breed, intramammary infection (IMI), which spreads within farms and pastures, can have a negative impact on animal health and, consequently, can cause relevant economic losses due to cow morbidity and mortality, reduced milk yield and quality, and increased antibiotic use [2]. Among major mastitis-causing pathogens, *Staphylococcus* (*S.*) *aureus* is linked to particular pathogenic and contagious properties, through which the cure rate is still low [3]. The diffusion and persistence of specific genotypes in dairy herds are associated with virulence-related genes [4,5], whose combinations influence the development of IMI. Therefore, molecular subtyping of *S. aureus* isolates is essential for assessing their pathogenic properties and guiding effective treatment strategies. Molecular tools such as Multi Locus Sequence Typing (MLST) [6] and Ribosomal spacer (RS)-PCR allow for the accurate discrimination of *S. aureus* strains [5,7], supporting treatment choices based on the antimicrobial susceptibility and pathogenicity of the infecting strain [8]. Given that *S. aureus* genetic diversity is influenced by geographical factors [9], understanding its distribution and resistance profiles in alpine pastures is crucial for effective disease control.

This study aims to investigate the correlation between *S. aureus* genotypes, antimicrobial resistance (AMR) patterns, and virulence profiles in Aosta Valley cattle breeds raised in alpine pastures. By characterizing strain dissemination and antimicrobial susceptibility, this research seeks to provide valuable insights into pathogen dynamics and contribute to the development of targeted mastitis control strategies.

## 2. Results

### 2.1. Bacteriological Results

A schematic overview of the study workflow is shown in Figure 1. Among the 468 milk samples analyzed, 386 (82.5%) were culture-positive, 74 (15.8%) were negative, and the remaining 8 (1.7%) were contaminated. Of the 386 positive cows, 244 (63.2%) had a single pathogen, of which 40 (16.4%) were infected by *S. aureus*, whereas 142 (36.8%) were co-infected, of which 47 (33.1%) were co-infected with *S. aureus* (Table 1). All four herds presented a high prevalence of *S. aureus* across all sampling points (T1: 21.6%; T2: 16.1%; T3: 15.3%).

### 2.2. Genotyping

#### 2.2.1. RS-PCR

Among the 87 *S. aureus* isolates from both single and co-infection, 82 were successfully genotyped (Table 2). The RS-PCR analysis revealed the presence of two major clusters (CLR and CLB), accounting for 86.6% of strains analyzed (Appendix A), except for farm 1 (Appendix A), in which only one cluster circulated, more than two clusters coexisted in the remaining herds. As reported in Table 2, the most prevalent and common genotype was GTR^I^ (61%), followed by GTB (14.6%). The GTR^I^ was detected in all four herds throughout the entire experimental period and was always predominant, whereas the GTB was found in only farms 2 and 3 (Appendix A). In addition to GTR^I^, cluster R revealed the presence of GTR (2.4%), GTR^IV^ (4.9%), and GTR^XIII^ (2.4%), whereas GTB^IV^ was the only variant for cluster B (1.2%). The remaining *S. aureus* strains (13.4%) belonged to minor genotypic profiles, each of which resulted from the analysis of 5 (GTBI), 2 (GTC), or 1 isolate (GTA^II^, GTZ, GTAP, GTJ^II^). Furthermore, the genotype distribution of *S. aureus* in different herds at different timepoints is presented in Appendix A.

#### 2.2.2. MLST

All the GTB and its variant strains analyzed in this study belonged to CC8-ST8. The MLST analysis of 15 GTR^I^ strains, chosen from the four different herds, revealed the full sequence identity of the seven genes considered, classifying them as ST97-CC97.

### 2.3. Antimicrobial Susceptibility

The phenotypic results obtained by MIC assay (Table 3) of the 82 successfully genotyped *S. aureus* strains revealed the presence of high levels of resistance, mainly to the natural penicillins and aminopenicillins. Indeed, 57 (69.5%) isolates were resistant to penicillin, 55 (67.1%) to ampicillin, and 42 (51.2%) to amoxicillin plus clavulanate, while no resistant isolate was found to cephalosporins and oxacillin. Among the non-β-lactam antibiotics, negligible resistance percentages were found against erythromycin (3.7%), enrofloxacin (2.4%), trimethoprim plus sulfamethoxazole (2.4%), and both aminoglycosides (2.4% and 4.9% for gentamicin and kanamycin, respectively). A greater number of isolates (9.75%) were resistant only to rifampin. The detection of non-wild type (non-WT) isolates by using the epidemiological cut-off (ECOFF) values revealed the coincidence of many ECOFFs with the clinical breakpoint (CBP) values, providing similar or equal results for β-lactam antibiotics, and the presence of only one (1.72%) non-WT isolate for cefazolin. Only a few ECOFF values were available for non-β-lactam antibiotics, and all of them were lower than the respective CBP values; nevertheless, the number of non-WT strains was equal to that of the resistant isolates for erythromycin and trimethoprim plus sulfamethoxazole, and slightly higher for gentamicin (four non-WT vs. two resistant isolates). Regarding gentamicin, it should be highlighted that the number of non-WT could be underestimated because the ECOFF is 1 µg/mL, and the first dilution of the MIC plate is 2 µg/mL.

### 2.4. Distribution of Antimicrobial Resistance Profiles Among Genotypic Clusters

Analysing the resistance to multiple classes of antimicrobials, 25.6% of the isolates were susceptible or intermediate to all the tested antimicrobials. Two (2.4%) isolates, one GTR^I^ and one GTBI, were resistant only to benzylpenicillins, while one (1.2%) isolate belonging to GTR was resistant only to macrolides. Most (53; 64.6%) *S. aureus* isolates were resistant to two different antimicrobial classes, namely aminopenicillins in addition to benzylpenicillins. The MDR rate was 6.1%, with two GTR^I^ strains showing resistance to rifamycins in addition to aminopenicillins and benzylpenicillins; one GTR strain was classified as resistant to macrolides, aminoglycosides, rifamycins and one GTZ strain to quinolones, aminoglycosides, and sulfonamides. In addition, one strain belonging to GTA^II^ was classified as resistant to quinolones, macrolides, aminoglycosides, and sulfonamides.

Comparing the MIC distributions of the two main CLs, Table 3 shows the differences between CLR and CLB. The MIC distribution of CLB was significantly lower compared to CLR for amoxicillin plus clavulanate (*p*-value = 0.000001), ampicillin (*p*-value = 0.000000003), cefazolin (*p*-value = 0.00001), penicillin (*p*-value = 0.00000002), and trimethoprim plus sulfamethoxazole (*p*-value = 0.0000001), while a slightly higher but still significant *p*-value was found for ceftiofur (*p*-value = 0.009) and oxacillin (*p*-value = 0.002). Figure 2 displays the bimodal MIC distributions for penicillin and ampicillin; the majority of the CLB strains had lower MIC values for ampicillin and penicillin.

### 2.5. Virulence and Antimicrobial Profiling

Out of 82 strains successfully genotyped, 71 were positive for the two major clusters (CLR and CLB). As 24 CLR and 5 CLB strains were isolated from milk samples of cows lost to slaughter, involuntary or biological culling, or sold, the remaining 42 strains (33 from CLR and 9 from CLB) isolated from milk samples of animals collected at all three timepoints were included in a thorough analysis (Table 4). Among the genes related to host invasion, *lukED* was detected in all of them without any difference between clusters or genotypes, while none carried the *lukE*, *scn* and *sak* genes. The *hla* (97.0%) and *hlb* (75.8%) genes were present in most CLR strains, whereas the *chp* (12.1%) and *fmtB* (54.4%) genes were present only in the GTR^I^. Only one GTR^I^ strain harbored *lukM* and a combination of two enterotoxin genes (*seg* and *sei*), while the remaining isolates were not enterotoxigenic and were characterized by absence of the other genes interfering with host defense (*tsst*, *eta* and *etb*). All the CLB strains were positive for the *hla* gene and negative for the enterotoxin genes, *chp* and *lukM*. Most of them were also positive for the combination of *fmtB* and *hlb* (88.9%); *cna* was detected in only 55.6% of the GTB strains. All the CLB strains were *adlb*-positive (8 with GTB and 1 with GTB^IV^), whereas CLR did not harbor the gene. Among the CLR strains, all the GTR^I^ strains were positive for genes responsible for biofilm formation (*icaA, icaB, icaC,* and *icaD*), whereas the remaining CLR variants (GTR and GTR^XIII^) and CLB (GTB and GTB^IV^) strains did not harbor these genes. Regarding AMR genes, all the strains were negative for *mecA*, *mecC*, *ermA, ermB* and *ermC*, regardless of genotypes. Most (93.9%) of the CLR strains (84.8% of GTR^I^ and 6.1% of GTR^XIII^) and half of the GTB strains (55.6%) harbored the *blaZ* gene.

### 2.6. Association Between Phenotypic Resistance and Resistance Genes 

The AMR phenotypic and genotypic patterns detected agreed for all the genes tested except *blaZ*. Among the 36 isolates that harboured the *blaZ* gene, only 28 were phenotypically resistant to penicillin. The 8 *blaZ* positive but penicillin susceptible isolates were tested for β-lactamase activity and did not demonstrate production of β-lactamase. 

## 3. Discussion

The use of RS-PCR confirmed the high frequency of clusters R and B, which predominated in Europe, together with CLC, CLF, and CLI [13]. In contrast to previous studies [5], the IMI rates of the four farms were strictly dependent on the most present CLR (70.7%), alongside the CLB (15.9%). The CLB has been previously identified as consistently involved in high within-herd IMI prevalence in the middle-south of Europe [14] due to its high contagiousness [15]. On the other hand, genotype R was combined with its 13 variants into a large cluster reported as “dairy cattle specific”, whose dissemination was likely initiated with the diffusion of cattle breeding from Europe to other countries long ago [9]. The variant GTR^I^ (61%), previously associated with low within-herd prevalence and restricted to a few infected cows [16], was predominantly represented in all four herds investigated in our study. The MLST method was employed in this work to further subtype *S. aureus* strains, allowing the identification of clones and determination of the evolutionary identity [17]. This nucleotide-based approach revealed that all the GTR^I^ strains belonged to CC97, known to be one *S. aureus* lineage strongly linked to cattle [18,19,20], whereas the GTB and its variants were exclusively recognized as the most pathogenic and virulent CC8-ST8 [21], which was found in bovine IMIs due to the recent bovine adaptation of this cluster as a consequence of a new human-to-cow host jump.

As in Lombardy [22], the bovine-associated CC8 and CC97 strains were the most frequent among methicillin-susceptible *S. aureus* (MSSA) detected, while no livestock-associated CC97 methicillin-resistant *S. aureus* (MRSA) strains were found in this study. Despite the growing concerns over *S. aureus* methicillin resistance emergence, the absence of *mec*A and *mec*C genes responsible for a decreased affinity for β-lactams [23,24,25] confirmed the low MRSA prevalence among *S. aureus* strains isolated from Italy [26] as well as in other European and global studies [9]. Oppositely, the phenotypic data revealed that the majority (69.5%) of the isolates were resistant to benzylpenicillins, probably resulting from the wide use of penicillin as the drug of choice over the past five decades [27,28,29]. The most significant resistance found in the Aosta Valley region was consistent with the percentage of penicillin-resistant *S. aureus* isolates from throughout Italy (58.8%) [30] but higher than in other European [31,32], Canadian and American countries [33,34]. The present study confirmed that penicillin resistance is often related to the genotypic cluster [32], with a frequency of 89.5% in CLR and 10.6% in CLB strains in this study. The high penicillin resistance rate found in CC97 was in line with previous results [32,35] but was linked to a lack of susceptibility to clavulanate. Out of 57 penicillin-resistant MSSA isolates, 42 were not susceptible to amoxicillin–clavulanate, confirming the noted resistance of MSSA isolates from human nasal samples to commonly used antibiotics [36,37]. Although penicillinase production should confer penicillin but not β-lactam plus clavulanate resistance, evidence from human medicine suggested that amoxicillin–clavulanate exposure in children could be associated with penicillinase hyperproduction by penicillin-resistant MSSA [38] and, consequently, with a reduction in the beta-lactamase inhibitors efficacy. The use of the amoxicillin–clavulanate combination in the considered herds could have induced a selection of isolates producing high enough levels of penicillinase to confer clinical resistance to this antimicrobial association, with an impact on *S. aureus* survival in its ecological niche. Although the investigated subset of strains was limited in our study, the combination of phenotypic and molecular methods revealed a disagreement on penicillin resistance assessment. Indeed, only 28 out of 36 isolates harboring the *blaZ* encoding β-lactamase [39] showed phenotypic resistance to penicillin, while 8 (66.7%) were phenotypically susceptible to penicillin. Similar results were reported in previous studies [30,40], and they raised the question of whether these isolates should be considered resistant to penicillin in any way. Haveri et al. suggested that the disagreement between phenotypic and molecular methods could arise from the use of incorrect resistance breakpoints [41]. The use of CBPs in veterinary medicine could be misleading due to the lack of CBPs for each combination of species–microorganism–disease, often necessitating laboratories to adapt CBPs from humans or other species. To overcome this issue, a double assessment of AMR using ECOFFs could be highly beneficial to ensure a more reliable evaluation of the AMR. Of course, the classifications based on ECOFFs are not immediately related to those based on CBPs because an isolate identified as a non-wild type may still be clinically susceptible [42]. On the other hand, in the specific case of penicillin, the CBP for the category “resistant” was the same as the ECOFF; moreover, these eight isolates tested negative for β-lactamase production, and their MIC was at the lower limit of the distribution range. Another possible explanation is that the detection of genes does not necessarily implicate their expression because the presence of the gene provides the potential for β-lactamase production by the bacteria when exposed to the antimicrobial. For this reason, the isolates susceptible to penicillin but positive for *blaZ* should always be considered potentially resistant. The *blaZ* gene was detected in 93.9% of CLR and 55.6% of CLB *S. aureus* strains, and its spread could significantly influence the epidemiology and treatment outcomes of staphylococcal infections [43]. Furthermore, the absence of *erm* genes corresponded with *S. aureus* isolates with macrolide susceptibility, in agreement with previous results [40] reporting that no *S. aureus* isolates susceptible to erythromycin carried *erm*C, but not all strains positive for the gene showed resistance to the specific antimicrobial. The generally low macrolide (3.7%) resistance confirmed the limited spread of genes responsible for ribosomal alternations [44], such as the methylation of specific 23S rRNA residues, and the enduring efficiency of erythromycin in treating staphylococcal mastitis [30,32]. Overall, the majority (63.4%) of the isolates showed resistance only to two antimicrobial classes, and the MDR incidence was low (6.1%).

The average usage of antimicrobials in the farms enrolled in the study was quite different, as shown in Table 5. Some farms greatly overtake the median antimicrobial consumption reported for dairy herds in Italy in 2021 and 2022, with 2.63 and 2.47 defined daily doses per animal for Italy, respectively (DDDAit). In all the farms but one, dry cow therapy accounted for almost 50% of antimicrobial consumption. However, there was a greater variation in lactation therapy, with some herds accounting 1/3 of the DDDAit to in, while in others, it accounted for less than 10%. The AMR detected was almost exclusively limited to the penicillins, while many other antimicrobials were used, especially in dry cow therapy. These data highlight the widespread complexity of AMR, which is linked not only to the overuse of antimicrobial therapy but also to the epidemiological and pathogenic patterns of the bacteria involved. In this case, the ability of the *S. aureus* cluster CLR to spread in the cow population concurred heavily with the diffusion of the AMR among the herd. For this reason, besides the reduction in antimicrobial treatment, the implementation of biosecurity measures is an essential requirement to reduce the presence of AMR in herds.

Despite the low penicillin resistance of CC8 in accordance with earlier results [32], these strains were closely linked to the *adlb* gene, which is located in the GTB-specific staphylococcal cassette chromosome SCCgtb and encodes adhesion-like bovine protein [45]. Even though the *adlb* was previously found also in GTR^I^ and dairy farms with predominant CC8 or CC97 strains carrying this gene had a higher within-herd prevalence of *S. aureus* IMI [32], our study detected it only in 66.7% of the GTB strains, leaving unclear the role of *adlb* as a staphylococcal marker for contagiousness and high IMI prevalence. On the other hand, all the GTR^I^ strains isolated in our study possessed the genes responsible for biofilm formation, encoding proteins that produce polysaccharide intercellular adhesion (PIA) and mediate cell-to-cell adhesion [46]. This mechanism is mostly controlled by the intracellular adhesion cluster (*icaADBC*) at the *ica* locus. In particular, the *icaA* and *icaD* genes promote the bacterial ability to survive not only by evading host immune defenses but also by affecting *S. aureus* antimicrobial susceptibility [47] through interaction between their own produced extracellular matrix and the active molecules [48,49]. As previously reported [9], all 42 *S. aureus* strains investigated were positive for *hla*, *hlb,* or both genes encoding α- and β-hemolysins, essential for the host invasion by forming transmembrane pores [50,51], as well as for the *luk*ED encoding the bicomponent leukotoxin. On the contrary, only one GTR^I^ strain carried the *luk*M, and none carried the *luk*S-PV produced the leukocidin S/F-PV, also known as Panton–Valentine leucocidin (PVL), which is able to kill neutrophils, macrophages, and dendritic cells [52,53,54,55]. Among the virulence genes involved in binding to host cells and damaging host tissues, the most common was *clfA* (73.8%), followed by *cna* (50%), encoding the clumping factor and the collagen-binding protein, respectively [56,57]. These surface adhesins belonged to the microbial surface components recognizing adhesive matrix molecules (MSCRAMMs) [58,59]. Of the immune evasion cluster (IEC) genes interfering with host immunity [60,61], only four GTR^I^ strains were positive for the chemotaxis inhibitory protein (*chp*) gene, whereas none were positive for the staphylococcal complement inhibitor (*scn*) or the staphylokinase (*sak*) genes, or even the gene for enterotoxin A (*sea*). Among the staphylococcal enterotoxins, only one strain (2.4%) of GTR^I^ harbored a combination of two enterotoxin genes (*seg* and *sei*), while the remaining (97.6%) isolates were not enterotoxigenic and were characterized by the absence of the toxic shock syndrome toxin-1 (*tsst-1*).

## 4. Materials and Methods

### 4.1. Ethics Statements

The research protocol was part of an agreement by the Italian Ministry of Health (authorization n. 628/2016-PR and OPBA_34_2024) with the University of Milan, and methods were carried out in accordance with the approved guidelines. Milk samples used for the analyses were collected by veterinarians in accordance with specific hygiene requirements. Cows were not subjected to any invasive procedures.

### 4.2. Herd Selection

In Aosta Valley, the traditional husbandry system is based on small herds housed in closed barns located in the valley during the winter. In early spring, the cows are given temporary access to lowland pastures around the farms for diurnal grazing. The dairy herds are led up to mountain pastures from early June up to late September and leave to return back to their farms every autumn. Four dairy farms of Aosta Valley with Valdostana breed under a DHIA (Dairy Herd Improvement Association) control program were selected for this study based on convenience sampling, according to the availability and interest level of dairy farmer participation.

During the grazing season, the first herd was led up into a mountain pasture at 2100 m asl, the second herd grazed on the pasture at 2500 m asl, and the third and the fourth herds grazed at 1300 and 2287 m asl, respectively. At the beginning of the study, the herds’ size ranged from approximately 80 to 120 Valdostana cows, with a 305-day lactation yield varying from 3500 to 4500 kg. The milk quality data reported in 2022 were the following: fat ranged from 3.3 to 3.5%, protein from 3.2 to 3.6%, casein from 2.8 to 4.2%, and somatic cell count (SCC) from 70,000 to 561,000 cells/mL. Data concerning the use of antimicrobial treatment were extracted from reports of the ClassyFarm database (Appendix A), the surveillance tool of the Italian veterinary public health system for monitoring livestock farms (https://www.vetinfo.it/ (accessed on 12 December 2024)). The measuring unit used to assess antimicrobial consumption by ClassyFarm is the defined daily dose for animals according to the definition of the European Medicine Agency [62]. Antimicrobial usage varied greatly among the herds, ranging from a defined daily dose animal for Italy (DDDAit) of 13.89 to 0.5, but there was an overall reduction from 2021 to 2022 (Table 5). The most commonly used antimicrobials for udder treatment belonged to the class of aminoglycosides, rifamycins, and β-lactams, among which aminopenicillins (amoxicillin plus clavulanate) and benzylpenicillins (penicillin) were the most frequently employed on the farms.

### 4.3. Milk Sample Collection

A total of 468 composite (pool of all four bovine quarters) milk samples were collected from 4 different dairy herds at three timepoints (T1: mountain pastures system from July to August 2022; T2: closed barns system from October 2022 to June 2023; T3: mountain pastures system in August 2023), as reported in Figure 1. Among the 227 (100%) cows initially screened before dry-off (T1), 143 (56.5%) were sampled after calving (T2), and 98 (38.7%) were tested again before dry-off (T3). Of the remaining 129 cows missed at T2 or T3, 84 cows were excluded from T2 after being returned from mountain pastures due to slaughter (53 cows, 63.1%), biological culling (27 cows, 32.1%), or selling (4 cows, 4.8%). Additionally, 45 cows were slaughtered (28 cows, 62.2%), biologically culled (16 cows, 35.6%), or sold (1, 2.2%) before leaving for the next summer season (T3).

Milk samples were collected by veterinarians in 50 mL sterile vials labeled with the cow number after disinfecting teat ends and discarding the first streams of foremilk, following the National Mastitis Council (NMC) recommendations [63]. After sampling, they were immediately stored at −20 °C and, at the end of each timepoint, shipped to the Laboratorio di Malattie Infettive degli Animali (MiLab, Università degli Studi di Milano, Lodi, Italy) for milk microbiological analysis; wet ice was used to keep samples frozen during transportation.

### 4.4. Bacteriological Analysis and Staphylococcus Aureus Identification

Milk samples were thawed and gently mixed to obtain a homogeneous suspension. After mixing, 10 μL of each milk sample was plated onto blood agar plates containing 5% defibrinated sheep blood (Microbiol, Cagliari, Italy). Plates were incubated aerobically at 37 °C and examined after 24 and 48 h according to the NMC guidelines [63]. The detection of mastitis pathogens was performed using Matrix Assisted Laser Desorption Ionization—Time of Flight mass spectrometry (MALDI-TOF MS) [64], with the MALDI Biotyper System (Bruker Daltonik GmbH, Bremen, Germany), by applying the direct transfer method as previously described [65,66]. For each *S. aureus* positive cow, a representative colony identified as *S. aureus* was re-isolated and stored at −20 °C with the Cryobank beads system (Technical Service Consultant Ltd., Lancashire, UK) for further phenotypic and genotypic characterization.

### 4.5. Antimicrobial Susceptibility Testing (AST)

The antimicrobial susceptibility of the *S. aureus* isolates was assessed using the Minimum Inhibitory Concentration (MIC) method by the Istituto Zooprofilattico Sperimentale delle Venezie (IZSVe), part of the Italian network of the state veterinary laboratories. The MIC values of 12 antimicrobials (amoxicillin/clavulanate, ampicillin, cefazolin, ceftiofur, enrofloxacin, erythromycin, gentamicin, kanamycin, oxacillin, penicillin, rifampin, and trimethoprim/sulfamethoxazole), selected for their activity against cattle mastitis pathogens and their registration by the Italian Ministry of Health, were determined for *S. aureus* isolates using the broth dilution test following the Clinical and Laboratory Standards Institute (CLSI) standard VET01 [11]. The MIC assay was performed with a customized commercial microdilution MIC system, including MIC plates and Cation Adjusted Mueller-Hinton Broth (CAMHB), supplied by TREK Diagnostic Systems (East Grinstead, UK). The MIC plates were produced in accordance with ISO 13485:2016, and each MIC plate batch underwent quality control (QC) using *S. aureus* ATCC^®^ 29213, Enterococcus faecalis ATCC 29212^®^, and Escherichia coli ATCC^®^ 35218 by the company before release. Each new batch of MIC plates was tested before usage by the laboratory using *S. aureus* ATCC^®^ 29213 and Escherichia coli ATCC^®^ 35218. The antimicrobials, which included a QC range value at the lower or upper limit of the plate dilutions, were assessed for QC, excluding these dilutions. The isolates were thawed and streaked onto blood agar plates, and colonies from overnight growths were picked with a sterile loop and suspended in sterile saline produced by an internal IZSVe-certified service provider (CSP). The bacterial suspension was adjusted using a nephelometer (Biosan, Riga, Latvia) until an optical density (OD) of 0.5 nephelometric turbidity units (NTUs) on the McFarland scale was achieved. The suspension was transferred to CAMHB to provide an inoculum concentration of approximately 5 × 10^5^ CFU/mL, and 100 µL of the CAM-HB was transferred to each well of the plate. After incubation at 35 ± 2 °C for 16–20 h, bacterial growth was assessed using a Sensititre Vizion instrument (Thermo Scientific, Loughborough, UK), and the last concentration of antimicrobial that did not show turbidity or a deposit of cells at the bottom of the well was recorded. The MIC value of each isolate, expressed as µg/mL, was defined as the lowest concentration of the antimicrobial agent that completely inhibited growth after the incubation period. MIC test performance was monitored by the laboratory checking the MIC ranges of the *S. aureus* ATCC^®^ 29213 strain, according to CLSI VET 01S recommendations [10] for the following antimicrobials: amoxicillin plus clavulanate, ampicillin, cefazolin, ceftiofur, erythromycin, oxacillin, penicillin, trimethoprim/sulfamethoxazole. The results were interpreted as susceptible, intermediate, and resistant using the CLSI clinical breakpoints (CBPs) [10] or the Comitè de l’Antibiogramme de la Sociètè Française de Microbiologie (CASFM) CBPs (CASFM, 2024) when CLSI CBPs were not available. It should be highlighted that the use of CASFM CBPs is not limited to the use of specific CASFM methods for testing antimicrobial susceptibility. If CBPs were differentiated for disease and host species, mastitis and cattle CBPs were selected when present; otherwise, other animal species and human breakpoints were used (Table 3). The results were also assessed using the epidemiological cut-off values (ECOFFs) provided by the European Committee on Antimicrobial Susceptibility (EUCAST) for the antimicrobials that had available ECOFFs [67] to detect non-wild type (non-WT) isolates (Table 3). It should be pointed out that while CBPs classify MIC values into distinct classes of bacterial susceptibility based on clinical outcomes of antimicrobial treatment, ECOFFs are based on the mathematical analyses of observed MIC distributions to differentiate wild-type and non-WT isolates. The median MIC (MIC_50_) and the 90th percentile (MIC_90_) were calculated for each antimicrobial, and subsequently, multi-drug resistance (MDR) was assessed according to the definition of resistance to three or more antimicrobial classes [68].

### 4.6. Molecular Analysis

#### 4.6.1. DNA Extraction

Genomic DNA was extracted from the pure culture of stored *S. aureus* isolates using a DNeasy kit (QIAGEN, Valencia, CA, USA) as previously described [69]. The amount and quality of the DNA were measured using a NanoDrop ND-1000 spectrophotometer (Nano-Drop Technologies, Wilmington, DE, USA), and the DNA was stored at −20 °C until use.

#### 4.6.2. RS-PCR

All *S. aureus* isolates were genotyped by RS-PCR, a method that amplifies the 16S–23S rRNA intergenic spacer region, coupled with a miniaturized electrophoresis system (Agilent Technologies, Santa Clara, CA, USA), as previously described [5,9]. Genotypes (GTs) were determined according to the outline method [5,16], which was enhanced by calculating the Mahalanobis distance of informative peak sizes and comparing them to those of the prototype strains using the “Mahalanobis Distances of *S. aureus* Genotypes” software, in-house developed C# program [70]. For interpretation of the results, genotypic variants differing in one electrophoretic band from identified genotypes were indicated by superscripted Roman numerals after the genotype name (e.g., GTR^I^). For further simplification, genotypes and their variants were combined into clusters (CLs).

#### 4.6.3. MLST

Internal PCR fragments of 7 housekeeping genes were amplified using primers and protocols previously described [6]. The PCR products were then purified and sequenced by Microsynth (https://www.microsynth.com/ (accessed on 2 May 2024)). For each strain, the obtained chromatograms were then checked for quality using the Chromas 2.4.2 software (www.softpedia.com (accessed on 2 May 2024)). Clonal analysis of the STs was carried out using e-BURST, a web-based clustering algorithm (https://pubmlst.org/organisms/staphylococcus-aureus/ (accessed on 2 May 2024)), organizing MLST datasets into groups of related isolates and predicts the founding genotype of each CC.

#### 4.6.4. Standard PCR

We analyzed by PCR 42 *S. aureus* strains isolated from milk samples of cows collected at all three timepoints. We investigated 28 virulence-associated genes related to host invasion (*chp*, *clfA*, *cna*, *fmtB*, *hla*, *hlb*, *luk*M, *luk*ED, *luk*SF-PV, *scn*, *sak*) [18,71,72,73,74,75], interfering with host defense (*tsst*, *eta*, *etb*, enterotoxin genes from *sea* to *sel*) [5,71,75,76], and involved in biofilm formation (*icaA*, *icaD*, *icaB*, *icaC*) [77], as well as the occurrence of 6 AMR genes (*blaZ*, *mec*A, *mec*C, *erm*A, *erm*B, *erm*C) [73,78,79,80,81]. The DNA was amplified by using the primer sequences and following the PCR conditions as previously published. All amplified PCR fragments were visualized on 2% agarose gel electrophoresis (GellyPhor, Euroclone, Milan, Italy) and stained with ethidium bromide (0.05 mg/mL; Sigma Aldrich, Milan, Italy) under UV transilluminator (BioView Ltd., Nes Ziona, Israel). A 100 bp DNA ladder (Finnzymes, Espoo, Finland) was included in each gel.

#### 4.6.5. Adlb-Targeted PCR

The *adlb*-targeted real-time PCR was performed on the RS-PCR genotyped isolates [45].

### 4.7. β-Lactamase Detection

The *S. aureus* isolates that were phenotypically susceptible to penicillin according to the MIC but positive for the *blaZ* gene were tested for phenotypic β-lactamase activity using the nitrocefin-based test (nitrocefin disks, Sigma-Aldrich, St. Louis, MO, USA), according to the manufacturer’s instructions. *S. aureus* ATCC 29213 and *S. aureus* ATCC 25922 were used as positive and negative controls, respectively.

### 4.8. Statistical Analysis

Statistical analyses were performed using SPSS 27.0 Statistics (IBM, Armonk, NY, USA). Descriptive statistics of the MIC values in different clusters were expressed as a mean ± standard deviation (SD). The normality of the distribution of MIC values was assessed by the Shapiro-Wilk test. Since the data were not normally distributed, the comparisons of MIC values in different clusters were assessed using a non-parametric test (Mann–Whitney U test) for 2 independent samples. *p*-values < 0.01 were considered statistically significant.

## 5. Conclusions

This study provides a comprehensive analysis of *S. aureus* genotypes, AMR, and virulence profiles in dairy herds in the Aosta Valley. The high prevalence of GTR^I^ and their molecular characteristics emphasize the importance of *S. aureus* typing in understanding transmission dynamics. In particular, the identification of key genes, such as *clfA*, *cna*, *lukED*, *fmtB*, *hla*, *hlb*, and those involved in biofilm formation, highlights the pathogenic potential of the predominant genotype. Although the overall MDR rate was low, the large presence of the *blaZ* gene in CLR and, consequently, the strong association between penicillin resistance and this genotypic cluster indicate the need for ongoing surveillance and targeted interventions. Our findings highlight the benefits of continuous monitoring of the diffusion of *S. aureus* genotypes with respect to their virulence factors and antimicrobial susceptibility. Further knowledge of *S. aureus* IMI epidemiology can help dairy farmers develop effective strategies to enhance the sustainability of alpine grazing systems in the Aosta Valley.

## Figures and Tables

**Figure 1 antibiotics-14-00348-f001:**
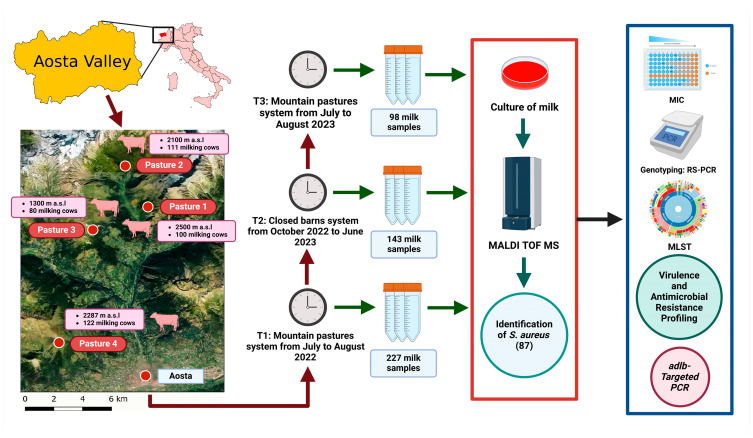
Overview of the study design. Figure created with BioRender.com.

**Figure 2 antibiotics-14-00348-f002:**
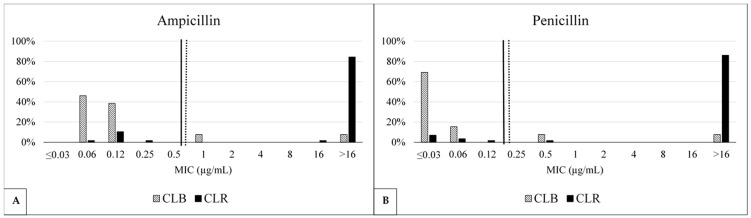
Comparison between (**A**) ampicillin and (**B**) penicillin minimum inhibitory concentration (MIC) distributions and the frequency of CLB (13) and CLR (58) strains. The MIC values and the percentages of *S. aureus* strains are displayed on the *x*-axis and the *y*-axis, respectively. The solid black and dashed lines represent the CLSI clinical breakpoint (CBP) value and the EUCAST epidemiological cut-off (ECOFF), respectively.

**Table 1 antibiotics-14-00348-t001:** Distribution of *S. aureus* and other bacterial intramammary infections over time.

Result	T1	T2	T3	Total
*S. aureus* (single infection)	24 (10.6%)	6 (4.2%)	10 (10.2%)	40 (8.6%)
*S. aureus* (co-infection)	25 (11%)	17 (11.9%)	5 (5.1%)	47 (10%)
Others ^a^	126 (55.5%)	104 (72.7%)	69 (70.4%)	299 (63.9%)
Contamination	0 (0%)	0 (0%)	8 (8.2%)	8 (1.7%)
Negative	52 (22.9%)	16 (11.2%)	6 (6.1%)	74 (15.8%)
Total	227 (100%)	143 (100%)	98 (100%)	468 (100%)

^a^ *Streptococcus* spp. and streptococci-like organisms, Non-Aureus Staphylococci and Mammaliicocci (NASM), Corynebacterium spp.

**Table 2 antibiotics-14-00348-t002:** Overall distribution of RS-PCR genotypes and genotypic clusters of *S. aureus* from cow milk samples collected across the 4 farms, including all strains at least isolated at one timepoint.

Cluster	Genotype	Number of Isolates (%)
CLR	GTR^I^	50 (61.0)
	GTR^XIII^	2 (2.4)
	GTR^IV^	4 (4.9)
	GTR	2 (2.4)
	Total	58 (70.7)
CLB	GTB	12 (14.6)
	GTB^IV^	1 (1.2)
	Total	13 (15.9)
CLBI	GTBI	5 (6.1)
CLA	GTA^II^	1 (1.2)
CLZ	GTZ	1 (1.2)
CLC	GTC	2 (2.4)
CLAP	GTAP	1 (1.2)
CLJ	GTJ^I^	1 (1.2)

**Table 3 antibiotics-14-00348-t003:** For each antimicrobial, the dilution range, the EUCAST epidemiological cut-off (ECOFF) and tentative ECOFF ((T)ECOFF) value, the percentage of *S. aureus* isolates with phenotypically detectable acquired resistance mechanisms (non-WT; non-wild type), the CLSI clinical breakpoint (CBP) value for the species indicated, the percentage of resistant (R) *S. aureus* isolates, and the MIC inhibiting the growth of at least 50% (MIC_50_) and 90% (MIC_90_) of the 82 *S. aureus* isolates analyzed.

Antimicrobial Class	Antimicrobial (µg/mL)	(T)ECOFF (μg/mL)	Non-WT (%)	CBP (μg/mL)	References and Species	Isolate (n)	R (%)	MIC_50_ (μg/mL)	MIC_90_ (μg/mL)
Aminopenicillins	Amoxicillin + clavulanate	>0.5/0.25	0	≥1/0.5	[10] Dog	CLB (13)	0	≤0.12/0.06	0.5/0.25
			68.97			CLR (58)	68.97	1/0.5	1/0.5
	(0.12/0.06–32/16)		51.22			All (82)	51.22	1/0.5	1/0.5
	Ampicillin	>0.5	15.38	≥1	[10] Horse	CLB (13)	15.38	0.12	1
			86.21			CLR (58)	86.21	>16	>16
	(0.03–16)		67.07			All (82)	67.07	>16	>16
1st generation Cephalosporins	Cefazolin	>2	0	≥8	[10] Dog	CLB (13)	0	0.5	0.5
			1.72			CLR (58)	0	1	1
	(0.12–8)		1.22			All (82)	0	1	1
3rd generation Cephalosporins	Ceftiofur	na ^a^	-	≥8	[10] Cattle	CLB (13)	0	1	1
			-			CLR (58)	0	1	2
	(0.12–32)		-			All (82)	0	1	2
Quinolones	Enrofloxacin	na ^a^	-	≥4	[10,11] Cat	CLB (13)	0	0.25	0.25
			-			CLR (58)	0	≤0.12	0.25
	(0.12–4)		-			All (82)	2.44	≤0.12	0.25
Macrolides	Erythromycin	>1	0	≥8	[10] Human	CLB (13)	0	0.25	0.25
			3.45			CLR (58)	3.45	≤0.12	≤0.12
	(0.12–8)		3.66			All (82)	3.66	≤0.12	≤0.12
Aminoglycosides	Gentamicin	>2	0	≥16	[10] Human	CLB (13)	0	≤2	≤2
			3.45			CLR (58)	0	≤2	≤2
	(2–32)		4.88			All (82)	2.44	≤2	≤2
	Kanamycin	na ^a^	-	≥16	[12] ^b^	CLB (13)	0	≤4	≤4
			-			CLR (58)	3.75	≤4	≤4
	(4–32)		-			All (82)	4.88	≤4	≤4
Antistaphylococcal penicillins	Oxacillin	>2	0	≥4	[10] Human	CLB (13)	0	0.5	0.5
			0			CLR (58)	0	1	1
	(0.12–4)		0			All (82)	0	1	1
Benzylpenicillins	Penicillin	>0.12	15.38	≥0.25	[10] Human	CLB (13)	15.38	≤0.03	0.5
			87.93			CLR (58)	87.93	>16	>16
	(0.03–16)		69.51			All (82)	69.51	>16	>16
Rifamycins	Rifampin	na ^a^	-	≥0.12	[12] ^b^	CLB (13)	7.60	≤0.06	≤0.06
			-			CLR (58)	10.34	≤0.06	0.12
	(0.06–2)		-			All (82)	9.75	≤0.06	≤0.06
Sulfonamides	Trimethoprim + sulfamethoxazole	>0.25/4.75	0	≥4/76	[10] Human	CLB (13)	0	0.25/4.75	0.25/4.75
			0			CLR (58)	0	0.12/2.37	0.12/2.37
	(0.12/2.37–4/76)		2.44			All (82)	2.44	0.12/2.37	0.25/4.75

^a^ Not available. ^b^ CASFM CBs are reported without species indication.

**Table 4 antibiotics-14-00348-t004:** Distribution of the main antimicrobial resistance and virulence genes between the two main *S. aureus* genotypes (GTR^I^ and GTB) only of the strains isolated from milk samples collected at all three timepoints.

Isolate (n)	Antimicrobial Resistance		Virulence Genes														Biofilm Formation							
	*blaZ*		*clfA*		*lukE-lukD*		*chp*		*fmtB*		*cna*		*hla*		*hlb*		*icaA*		*icaB*		*icaC*		*icaD*	
	n	%	n	%	n	%	n	%	n	%	n	%	n	%	n	%	n	%	n	%	n	%	n	%
GTR^I^ (30)	28	93.3	22	78.6	30	100	4	13.3	18	60	14	46.7	29	96.7	23	76.7	30	100	30	100	30	100	30	100
GTB (8)	5	62.5	7	87.5	8	100	0	0	7	62.5	5	55.6	8	100	7	87.5	0	0	0	0	0	0	0	0

**Table 5 antibiotics-14-00348-t005:** Average parity, somatic cell count, and defined daily doses of antimicrobial agents (total and by administration route) across the four farms.

Year	2022	2022	2021 2022	2021 2022	2021 2022
Farm	Parity (n)	SCC ^a^ (cells/mL)	DDDAit ^b^	IMM ^c^ LCT ^d^ DDDAit ^b^	IMM ^c^ DCT ^e^ DDDAit ^b^
1	3.6	70.000	3.78 1.47	1.63 0.66	1.84 0.16
2	2.8	409.000	2.34 2.14	0.69 0.83	0.0 0.0
3	3.4	124.000	13.89 5.61	3.29 2.84	5.03 0.25
4A ^f^	4.2	561.000	6.50 0.51	0.33 0.00	3.98 0.0
4B ^f^	4.2	561.000	3.03 3.54	0.20 0.00	1.57 1.03

^a^ Somatic cell count; ^b^ defined daily dose animal for Italy; dose of active agents expressed in mg/kg/day; ^c^ intramammary; ^d^ lactating cow therapy; ^e^ dry cow therapy; ^f^ a single dairy herd located in two farms with different business names.

## Data Availability

The data will be available upon request from the first authors (Valentina Monistero and Delower Hossain) and the corresponding author (Paolo Moroni).

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
