# Peer review of "Prevalence of Variant GTRIStaphylococcus aureus Isolated from Dairy Cow Milk Samples in the Alpine Grazing System of the Aosta Valley and Its Association with AMR and Virulence Profiles"

_antibiotics, 2025, doi:10.3390/antibiotics14040348_

Round 1
Reviewer 1 Report
Comments and Suggestions for Authors
Title: The title should contain the information that the isolates are from milk samples of cattle.
Line 73: (S.) – round brackets not in italics, only S. in italics
Line 98: Please explain the abbreviation NASM
Line 103: The closing bracket must not be in italics
Line 104 ff: What does CLR, CLB and the other cluster abbreviations stand for? Do the cluster designations follow a harmonized nomenclature or have they been determined by the authors. The same question arises for the genotype designations.
Lines 116-119: What about the MLST types of the few other genotypes as well as the GTR, GTRXIII and GTRIV isolates
Line 131, 139, Table 3: Please correct gentamicin
Line 132: Please delete the first “resistant”
Line 134: What does CBPs stand for? If clinical breakpoints are meant, clinical breakpoints are not resistant, but allow the classification of an isolate as resistant.
Lines 139-140: either non-WT or NWT, but not two abbreviations for the same classification
Lines 139-141: Such a plate should either not be used or at least gentamicin should not be evaluated. Please remember that the quality control range for S. aureus ATCC 29213 for gentamicin is 0.12-1 mg/L, which means that the results obtained for gentamicin cannot be validated by using this plate.
Table 3: ug must read µg; specie must read species; Please specify that the clinical breakpoint used are for the animal species indicated.
Table 3: The results have been evaluated according clinical breakpoints from different performance standards. The authors should be aware that a performance standard and the corresponding clinical breakpoints represent an entity. If the testing is performed according to CLSI recommendations, the results cannot be evaluated according to CASFM breakpoints (also lines 383-384) … and vice versa. For enrofloxacin ref. 11 is indicated – however, to the best of my knowledge ref 11 (CLSI document VET01) does not contain any clinical breakpoints. They are only in the CLSI document VET01S (ref 10).
Table 3: In general, the results obtained and presented in Table 3 must be used with caution. For only two of the antimicrobial agents tested, clinical breakpoints approved for bovine mastitis pathogens are available. Evaluating the MIC data by using clinical breakpoint approved for humans, cats, dogs or horses is problematic.
Table 3: The composition of the microtiter plate used for obtaining the MIC data is highly problematic. Besides gentamicin as mentioned before, the QC ranges of enrofloxacin (0.03-0.12 mg/L), kanamycin (1-4 mg/L), pirlimycin (0.12-1 mg/L) and rifampicin (0.004-0.015 mg/L) are below the corresponding test ranges on the plate (enrofloxacin: 0.12 – 4 mg/L, kanamycin: 4-32, pirlimycin: 2-4 mg/L; rifampicin: 0.06-2 mg/L). Moreover, the lower end of the QC ranges are the same as the lowest test concentrations for amoxicillin + clavulanate and oxacillin. As a consequence, the complete QC range cannot be determined. If the QC strain does not grow in the lowest test concentration, its MIC can be at the lowest test concentration, but also lower (in that case, it is out of range). The general rule says that for proper QC, at least one test concentration below the lowest concentration in the QC range should be on the microtiter plate. The authors may have a look at the recently published report: doi: 10.1093/jac/dkae471
Line 161: “not susceptible” does not necessarily mean “resistant” – “intermediate” does not count for MDR.
Lines 233-234: … the CB for the category “resistant” was the same as the ECOFF.
Line 237: blaZ in italics
Line 243: erm(C) with erm in italics and (C) not in italics
Line 248: “… ribosomal modifications.” The erm genes methylate specific residues in the 23S rRNA.
Line 262: icaADBC as well as ica in italics
Line 263: the same for icaA, icaD
In general, all genes must be in italics, including hla, hlb (line 267), lukED (268), lukM, lukSPV (269), clfA, cna (273), chp (277), scn (278), sak, sea (279), seg, sei (281), tst-1 (282) – the gene for toxic shock syndrome toxin 1 is name tst-1, not tsst-1
Line 322: What is a composite milk sample?
Line 362: The CLSI document VET01 is a standard, not a guideline.
Lines 373, 374, 504: CAMHB, not CAM-HB
References: Bacterial genus and species names must be in italics, aureus must be written with a lower case “a”
Ref 29: the five last authors are missing
Ref 80 and 82: Please correct the writing of the gene designations
Journal names should be abbreviated as in PubMed – some are abbreviated others not, please harmonize.
Author Response
Title: The title should contain the information that the isolates are from milk samples of cattle.
AU: a new title has been inserted with the asked information
Line 73: (S.) – round brackets not in italics, only S. in italics
AU: corrected as suggested
Line 98: Please explain the abbreviation NASM
AU: corrected as suggested
Line 103: The closing bracket must not be in italics
AU: corrected as suggested
Line 104 ff: What does CLR, CLB and the other cluster abbreviations stand for? Do the cluster designations follow a harmonized nomenclature or have they been determined by the authors? The same question arises for the genotype designations.
AU: CLR and CLB are abbreviations used to designate specific genotype clusters. Genotypes are named sequentially from GTA to GTZ, followed by GTAA to GTAZ, GTBA to GTBZ, and so on. Genotypic variants, which differ by only one band after electrophoretic analysis, are indicated with Roman numeral superscripts (e.g., GTRI). RS-PCR genotypes can be further grouped into clusters (CL), each of which consists of a genotype and all its variants (e.g., GTR and GTRI, GTRII, GTRIII, etc.)
The cluster and genotype nomenclature follows a system described in an available database, based on the RS-PCR profile of S. aureus strains. However, there is no internationally harmonized nomenclature, as the cluster names were determined by the authors based on the analysis of experimental data. Further details on the RS-PCR method and classification system can be found in Graber HU (Graber HU. Genotyping of Staphylococcus aureus by Ribosomal Spacer PCR (RS-PCR). J Vis Exp. 2016 Nov 4;(117):54623. doi: 10.3791/54623), which provides a comprehensive explanation of the genotyping process.
Lines 116-119: What about the MLST types of the few other genotypes as well as the GTR, GTRXIII and GTRIV isolates
AU: In this study, we analyzed the most common RS-PCR profiles (GTRI and GTB) using different methods, as these were the ones we aimed to investigate in more detail. Therefore, MLST was performed only on a subset of CLR and CLB strains. The results for GTR, GTRIV, and GTRXIII were CC130-ST3044, CC133-ST133, and CC352-ST352. We decided not to discuss these results because we did not analyze the minority clusters (from CLBI to CLJ in Table 2), considering this information not relevant, as our focus was on GTRI and GTB.
Line 131, 139, Table 3: Please correct gentamicin
AU: corrected as suggested
Line 132: Please delete the first “resistant”
AU: corrected as suggested
Line 134: What does CBPs stand for? If clinical breakpoints are meant, clinical breakpoints are not resistant, but allow the classification of an isolate as resistant.
AU: the word resistant was erased
Lines 139-140: either non-WT or NWT, but not two abbreviations for the same classification
AU: corrected using the abbreviation non-WT
Lines 139-141: Such a plate should either not be used or at least gentamicin should not be evaluated. Please remember that the quality control range for S. aureus ATCC 29213 for gentamicin is 0.12-1 mg/L, which means that the results obtained for gentamicin cannot be validated by using this plate.
AU: we responded to this comment at the fourth comment of table 3, where the same issue was raised also for other antimicrobials.
Table 3: ug must read µg; specie must read species; Please specify that the clinical breakpoint used are for the animal species indicated.
AU: corrected as suggested
Table 3: The results have been evaluated according clinical breakpoints from different performance standards. The authors should be aware that a performance standard and the corresponding clinical breakpoints represent an entity. If the testing is performed according to CLSI recommendations, the results cannot be evaluated according to CASFM breakpoints (also lines 383-384) … and vice versa. For enrofloxacin ref. 11 is indicated – however, to the best of my knowledge ref 11 (CLSI document VET01) does not contain any clinical breakpoints. They are only in the CLSI document VET01S (ref 10).
AU: We partially agree with this comment. The CLSI standard Vet08 suggest that CBs tables are valid only if the methodologies of standard VET01 are followed; therefore, the use of CLSI standard for MIC preparation is suggested when CLSI CBs, are used, but there are no limitations from other organizations as CASFM or EUCAST in the use of their CBP if the CLSI standard have been used for MIC preparation. The rationale of this absence of limitations lies in the need at European level to set CBPs according to the different therapeutical guidelines used, mainly in human medicine. In other words, the aim of EUCAST (SFM is part of the EUCAST organization) is to harmonize CB for existing or new antimicrobial agents in Europe, not to set an alternative standard for antimicrobial susceptibility test. Here two example of the use of CBs of different institutions:
Clin Microbiol Infect 2012; 18: 268–281 10.1111/j.1469-0691.2011.03570.x
Antibiotics 2017, 6, 39; doi:10.3390/antibiotics6040039
The above comment has been reported synthetically in the Material and method chapter.
Enrofloxacin reference 11 was erased as suggested
Table 3: In general, the results obtained and presented in Table 3 must be used with caution. For only two of the antimicrobial agents tested, clinical breakpoints approved for bovine mastitis pathogens are available. Evaluating the MIC data by using clinical breakpoint approved for humans, cats, dogs or horses is problematic.
AU: We agree with the reviewer about the issue posed by the use of CBPs in veterinary medicine. Unfortunately, due to the lack of CBs for each combination of specie - microorganism – disease, some CBPs should be adapted by other species, and this could provide unreliable results. We tried to overcome this problem using a double assessment based also on ECOFFs, which ensures a better comparability of data over time and facilitates the comparison of resistance between animal species (see EFSA Journal, https://doi.org/10.2903/j.efsa.2023.7867).
A specific comment about this issue was included both in chapter material and methods and discussion
Table 3: The composition of the microtiter plate used for obtaining the MIC data is highly problematic. Besides gentamicin as mentioned before, the QC ranges of enrofloxacin (0.03-0.12 mg/L), kanamycin (1-4 mg/L), pirlimycin (0.12-1 mg/L) and rifampicin (0.004-0.015 mg/L) are below the corresponding test ranges on the plate (enrofloxacin: 0.12 – 4 mg/L, kanamycin: 4-32, pirlimycin: 2-4 mg/L; rifampicin: 0.06-2 mg/L). Moreover, the lower end of the QC ranges are the same as the lowest test concentrations for amoxicillin + clavulanate and oxacillin. As a consequence, the complete QC range cannot be determined. If the QC strain does not grow in the lowest test concentration, its MIC can be at the lowest test concentration, but also lower (in that case, it is out of range). The general rule says that for proper QC, at least one test concentration below the lowest concentration in the QC range should be on the microtiter plate. The authors may have a look at the recently published report: doi: 10.1093/jac/dkae471
AU: We agree with the reviewer about the importance of testing the complete QC range of the antimicrobial included in the plate used for MIC assessment. The procedure we apply aim to cover QC ranges for at least one of the most frequently used CLSI-approved QC strains. Therefore, multiple strains are used to cover all QC ranges, to ensure that all the antimicrobials of the plate are tested. The QC procedure is performed at different level: the first is the quality control performed by the company, at the release of the plates, using multiple CLSI strains; the second one is performed at the laboratory using two strains, (S. aureus ATCC29213 E. coli ATCC 25922), to verify the compliance of the new batch of plates received; the third one is performed at each working session to assess the correct execution of the procedure. Only this last control is performed with a lower number of antimicrobials, because a single strain is tested (S. aureus ATCC29213). A more detailed description of QC procedure has been included in the Material and method chapter and, according to your suggestion, results of pirlimycin, the only antimicrobial that do not allow a complete QC of all the dilution range, have been erased form the manuscript.
Line 161: “not susceptible” does not necessarily mean “resistant” – “intermediate” does not count for MDR.
AU: Modified using the resistant instead of not susceptible
Lines 233-234: … the CB for the category “resistant” was the same as the ECOFF.
AU: : corrected as suggested
Line 237: blaZ in italics
AU: corrected as suggested
Line 243: erm(C) with erm in italics and (C) not in italics
AU: corrected as suggested
Line 248: “… ribosomal modifications.” The erm genes methylate specific residues in the 23S rRNA.
AU: the text has been corrected, accordingly.
Line 262: icaADBC as well as ica in italics
AU: corrected as suggested
Line 263: the same for icaA, icaD
AU: corrected as suggested
In general, all genes must be in italics, including hla, hlb (line 267), lukED (268), lukM, lukSPV (269), clfA, cna (273), chp (277), scn (278), sak, sea (279), seg, sei (281), tst-1 (282) – the gene for toxic shock syndrome toxin 1 is name tst-1, not tsst-1
AU: Sorry, we likely encountered an issue while organizing our paper according to the draft (due to the copy-and-paste function). We have corrected it as suggested
Line 322: What is a composite milk sample?
AU: Composite milk samples, in which milk from all four bovine quarters is collected into a single vial, are widely used in many developed dairy industries for detecting intramammary infections (IMI), as exemplified in the following references: 1) K. Reyher,I.R.Dohoo, 2011 Diagnosing intramammary infections: Evaluation of composite milk samples to detect intramammary infections.JDS Vol. 94, Issue 7 Pages 3213-3723; 2) Petzer IM, Karzis J, Donkin EF, Webb EC, Etter EM. Somatic cell count thresholds in composite and quarter milk samples as indicator of bovine intramammary infection status. Onderstepoort J Vet Res. 2017 Mar 24;84(1):e1-e10. doi: 10.4102/ojvr.v84i1.1269 3) Reydams H, Toledo-Silva B, Mertens K, Piepers S, de Souza FN, Haesebrouck F, De Vliegher S. Comparison of non-aureus staphylococcal and mammaliicoccal species found in both composite milk and bulk-tank milk samples of dairy cows collected in tandem. J Dairy Sci. 2023 Nov;106(11):7974-7990. doi: 10.3168/jds.2022-23092. In the text, the sentence has been corrected accordingly.
Line 362: The CLSI document VET01 is a standard, not a guideline.
AU: The sentence has been corrected, accordingly
Lines 373, 374, 504: CAMHB, not CAM-HB
AU: corrected as suggested
References: Bacterial genus and species names must be in italics, aureus must be written with a lower case “a”
AU: corrected as suggested
Ref 29: the five last authors are missing
AU: corrected as suggested
Ref 80 and 82: Please correct the writing of the gene designations
AU: corrected as suggested
Journal names should be abbreviated as in PubMed – some are abbreviated others not, please harmonize.
AU: corrected as suggested
Reviewer 2 Report
Comments and Suggestions for Authors
The manuscript written by Valentina Monistero et al analyzed the genotypes, antibiotic resistance and virulence profiles of staphylococcus aureus from alpine grazing system in the Aosta Valley. By charactering the strain dissemination, their study provides valuable perspectives to pathogen control strategies, which will benefit the livestock industry in the Aosta Valley region. However, the article presents certain issues regarding standardization and formatting, such as the notation of gene names (see minor comments below). It is recommended that the authors address these aspects during revision to enhance the rigor and readability of the manuscript. Additionally, the manuscript specifies the timeframe and farms from which strains were collected, as illustrated in Figure S1. Given this context, I would expect further discussion on how the authors perceive the impact of antimicrobial usage on the AMR profiles observed.
Minor:
- Abbreviations must be defined upon first use. Please clarify the full names of terms such as CLR, CLB, GTR, GTB, and AMR.
- Gene names should be italicized. Please correct all non-italicized gene names throughout the manuscript, including but not limited to those in lines 203–282, 427, 429, and 430.
- In Table 3, correct all instances of "ug/ml" to "μg/ml." Additionally, for CLR in erythromycin, if both MICâ‚…â‚€ and MIC₉₀ are lower than 0.12 μg/ml, how did the authors determine an R% of 3.45%? Please clarify.
- Line 200: Rephrase “while no livestock-associated CC97 methicillin-resistant S. aureus (MRSA) strains were not found in this study” for clarity. The current phrasing is confusing.
- Line 205: A comma is needed after “Oppositely.”
- Line 244: The phrase "in vitro" should be italicized.
Major:
- To improve readability, consider moving Figure 2 from the Methods section to the beginning of the Results section.
- In Table 3, under the columns for references and species, Citation 12 does not list a species. Additionally, this citation appears to be inaccessible—please verify its availability.
- Line 172: In the title of Figure 1, specify the number of CLB and CLR strains tested. Furthermore, the statement that "the majority of the CLB strains [are] at the lower end and most CLR strains [are] at the upper end" is imprecise. A more rigorous statement, such as "the majority of the CLB strains had lower MIC values for ampicillin and penicillin," would improve clarity.
- In Table 4, the authors tested 30 GTR1 strains and 8 GTB strains, whereas Table 2 indicates that 50 GTR1 strains and 12 GTB strains were isolated. Is there a reason why not all strains were tested? Additionally, the statement “Comparing the MIC distributions of the two main CLs, Table 4 showed the differences between CLR and CLB” seems to emphasize AMR gene testing. The authors should clarify their rationale for testing virulence and biofilm-associated genes and provide an explanation of their relevance.
- Line 430: The manuscript states that six AMR genes were tested, but only blaZ is shown in Table 4. What about the other genes? Please include them or clarify their omission.
Author Response
The manuscript written by Valentina Monistero et al analyzed the genotypes, antibiotic resistance and virulence profiles of staphylococcus aureus from alpine grazing system in the Aosta Valley. By charactering the strain dissemination, their study provides valuable perspectives to pathogen control strategies, which will benefit the livestock industry in the Aosta Valley region. However, the article presents certain issues regarding standardization and formatting, such as the notation of gene names (see minor comments below). It is recommended that the authors address these aspects during revision to enhance the rigor and readability of the manuscript. Additionally, the manuscript specifies the timeframe and farms from which strains were collected, as illustrated in Figure S1. Given this context, I would expect further discussion on how the authors perceive the impact of antimicrobial usage on the AMR profiles observed.
AU: we recognize the importance of provide a specific comment on the relation between antimicrobial usage in the herds and the AMR profiles observed. A specific comment about this outcome have been included in the chapter discussion
Minor:
Abbreviations must be defined upon first use. Please clarify the full names of terms such as CLR, CLB, GTR, GTB, and AMR.
AU: the text has been corrected, accordingly
Gene names should be italicized. Please correct all non-italicized gene names throughout the manuscript, including but not limited to those in lines 203–282, 427, 429, and 430.
AU: corrected as suggested
In Table 3, correct all instances of "ug/ml" to "μg/ml." Additionally, for CLR in erythromycin, if both MICâ‚…â‚€ and MIC₉₀ are lower than 0.12 μg/ml, how did the authors determine an R% of 3.45%? Please clarify.
AU: Among 58 isolates belonging to CLR group, 2 isolates (3.45%) were classified as resistant with a MIC value ≥ 8 μg/ml, 53 isolates (91.38%) had a MIC ≤ 0.12 μg/ml, 3 isolates had a MIC value equal to 0.25 μg/ml. Therefore, considering that more than 90% of isolates had a MIC ≤ 0.12 μg/ml, the lower dilution value, the value of MICâ‚…â‚€ and MIC₉₀ is equal or lower to 0.12. μg/ml
Line 200: Rephrase “while no livestock-associated CC97 methicillin-resistant S. aureus (MRSA) strains were not found in this study” for clarity. The current phrasing is confusing.
AU: corrected as suggested
Line 205: A comma is needed after “Oppositely.”
AU: corrected as suggested
Line 244: The phrase "in vitro" should be italicized.
AU: corrected as suggested
Major:
- To improve readability, consider moving Figure 2 from the Methods section to the beginning of the Results section.
AU: Thank you for your suggestion. We have moved Figure 2 from the Materials and Methods section to the beginning of the Results section to improve readability. We have also ensured consistency in the figure numbering and references throughout the text.
In Table 3, under the columns for references and species, Citation 12 does not list a species. Additionally, this citation appears to be inaccessible—please verify its availability.
AU: CASFM CBs are reported in the guideline without species indication. A footnote in table 3 was introduced for this explanation
Link: https://www.sfm-microbiologie.org/wp-content/uploads/2024/06/CASFM2024_V1.0.pdf
Line 172: In the title of Figure 1, specify the number of CLB and CLR strains tested. Furthermore, the statement that "the majority of the CLB strains [are] at the lower end and most CLR strains [are] at the upper end" is imprecise. A more rigorous statement, such as "the majority of the CLB strains had lower MIC values for ampicillin and penicillin," would improve clarity.
AU: The number of CLB and CLR strains tested has been added to the title of Figure 1. The text has also been revised accordingly.
In Table 4, the authors tested 30 GTR1 strains and 8 GTB strains, whereas Table 2 indicates that 50 GTR1 strains and 12 GTB strains were isolated. Is there a reason why not all strains were tested? Additionally, the statement “Comparing the MIC distributions of the two main CLs, Table 4 showed the differences between CLR and CLB” seems to emphasize AMR gene testing. The authors should clarify their rationale for testing virulence and biofilm-associated genes and provide an explanation of their relevance.
AU: Thank you for pointing out the discrepancy between Table 2 and Table 4. We would like to clarify this point. Table 2 presents the overall distribution of S. aureus genotypes and genotypic clusters across the four farms, including all strains isolated from the four dairy herds, at least at one timepoint. On the other hand, Table 4 shows the distribution of the main antimicrobial resistance and virulence genes between the two main genotypes (GTRI and GTB), but this analysis was based only on the S. aureus strains isolated from milk samples collected at all three timepoints, as described in the Materials and Methods section. Therefore, not all strains shown in Table 2 were included in Table 4, where we found only the strains that met the criteria for inclusion in the PCR analysis for resistance and virulence genes.
We hope this clarifies the reason for the difference in the number of strains tested between Table 2 and Table 4. We will make this explanation clearer in the Table 2 and 4 to avoid any further confusion.
Moreover, thank you for your comments, but we believe there may be some misunderstanding between Figure 1 and Table 4 and we would like to clarify this. The figure legend provides a detailed explanation of the data shown in Figure 1 above, while the caption of table explains the main antimicrobial resistance and virulence genes between the two main genotypes in the Table 4 below. The comparison of MIC distributions is actually shown in Figure 1, which focuses on the minimum inhibitory concentrations (MICs) of ampicillin and penicillin for the main genotypic clusters (CLR and CLB). Table 4 presents the distribution of antimicrobial resistance and virulence genes between the two main genotypes (GTRI and GTB), focusing on the genetic profiles not on the MIC values.
Line 430: The manuscript states that six AMR genes were tested, but only blaZ is shown in Table 4. What about the other genes? Please include them or clarify their omission.
AU: In Table 4, we reported the specific results for GTRI and GTB, the two main genotypes. The strains (from GTRI and GTB) analyzed for antimicrobial resistance genes (blaZ, mecA, mecC, ermA, ermB, ermC) tested positive only for blaZ, while all other targets were negative. Therefore, the results for the negative targets were not included in the table.
Round 2
Reviewer 1 Report
Comments and Suggestions for Authors
I still do not agree with your explanation of the interchangable use of clinical breakpoints from CLSI, CA-SFM and EUCAST. The current doctrine is that if you perform AST according to CLSI, you must use CLSI-approved breakpoints to evaluate the results. and this is also true for AST performance according to EUCAST and CA-SFM. The reason is simply that each organisation has set breakpoints based on their methodology ... and all the methodologies vary a bit from each other. However, I admit that there are published approaches to harmonize the use of breakpoints.
As you have used quite a number of QC strains, at least for the Enterococcus faecalis QC strain the test range and the QC range seem to fit for gentamicin and most of the other agents tested. Solely for kanamycin and rifampicin, it is a bit problematic. I wondered why these two agents have been tested. Both are not used in mastitis therapy.
Lines, 254, 457: There is a specifc nomenclature for MLS resistance genes. According to that, the genes erm(A), erm(B) and erm(C) are writen with erm in italics and A,B,C in round brackets and not in italics.
In all the gene designations, I would write the capital letters also in italics - this is not uniform in the manuscript and I suggest to harmonize that.
These latter two minor corrections can surely be done during the page proof corrections.
Reviewer 2 Report
Comments and Suggestions for Authors
Thanks for your response and revision.